# Prevalence of Cardiovascular Risk Factors and Coronary Angiographic Findings in High-Risk Immigrant Communities in Italy

**DOI:** 10.3390/jpm13060882

**Published:** 2023-05-23

**Authors:** Saverio Muscoli, Aikaterini Andreadi, Claudia Tamburro, Massimo Russo, Roberto Rosenfeld, Pietro Oro, Mihaela Ifrim, Federica Porzio, Lucy Barone, Francesco Barillà, Davide Lauro

**Affiliations:** 1Division of Cardiology, Fondazione Policlinico “Tor Vergata”, Viale Oxford 81, 00133 Rome, Italypietrooro94@gmail.com (P.O.); francesco.barilla@uniroma2.it (F.B.); 2Department of Systems Medicine, University of Rome “Tor Vergata”, 00133 Rome, Italy; andreadi@med.uniroma2.it (A.A.); roberto.rosenfeld88@gmail.com (R.R.);; 3Division of Endocrinology and Diabetology, Department of Medical Sciences, Fondazione Policlinico “Tor Vergata”, Viale Oxford 81, 00133 Rome, Italy

**Keywords:** coronary artery disease, ethnicity, South Asians, North Africa, Eastern Europe, Acute Coronary Syndrome, race

## Abstract

Background: The prevalence of coronary artery disease (CAD) considerably varies by ethnicity. High-risk populations include patients from Eastern Europe (EEP), the Middle East and North Africa (MENAP) and South Asia (SAP). Methods: This retrospective study aims to highlight cardiovascular risk factors and specific coronary findings in high-risk immigrant groups. We examined the medical records and coronary angiographies of 220 patients from the above-mentioned high-risk ethnic groups referred for Acute Coronary Syndrome (ACS) and compared them with 90 Italian patients (IP) from 2016 to 2021. In the context of high-risk immigrant populations, this retrospective study aims to shed light on cardiovascular risk factors and particular coronary findings. We analyzed the medical records of 220 patients from the high-risk ethnic groups described above referred for ACS and compared them with 90 IPs between 2016 and 2021. In addition, we assessed coronary angiographies with a focus on the culprit lesion, mainly evaluating multi-vessel and left main disease. Results: The mean age at the first event was 65.4 ± 10.2 years for IP, 49.8 ± 8.5 years for SAP (Relative Reduction (ReR) 30.7%), 51.9 ± 10.2 years for EEP (ReR 26%) and 56.7 ± 11.4 years for MENAP (ReR 15.3%); *p* < 0.0001. The IP group had a significantly higher prevalence of hypertension. EEP and MENAP had a lower prevalence of diabetes. EEP and MENAP had a higher prevalence of STEMI events; SAP showed a significant prevalence of left main artery disease (*p* = 0.026) and left anterior descending artery disease (*p* = 0.033) compared with other groups. In SAP, we detected a higher prevalence of three-vessel coronary artery disease in the age group 40–50. Conclusions: Our data suggest the existence of a potential coronary phenotype in several ethnicities, especially SAP, and understate the frequency of CV risk factors in other high-risk groups, supporting the role of a genetic influence in these communities.

## 1. Introduction

Coronary artery disease (CAD) is a major cause of morbidity and mortality worldwide and is an important medical and public health issue, and its prevalence widely varies by ethnicity [1].

CAD variability prompted the use of the Systematic Coronary Risk Evaluation 2 (SCORE2) and Systematic Coronary Risk Evaluation 2-older persons (SCORE 2-OP) risk charts in the most recent European Society of Cardiology cardiovascular (CV) prevention guidelines, which are optimized to four country groups (low, moderate, high and very high risk) based on national cardiovascular disease (CVD) mortality rates identified by the World Health Organization (WHO) [2,3].

High- and very high-risk regions include countries in Eastern Europe and North Africa, such as Albania, Bosnia and Herzegovina, Croatia, the Czech Republic, Hungary, Kazakhstan, Poland, Algeria, Azerbaijan, Egypt, Georgia, Morocco, Romania, the Russian Federation, Tunisia and Ukraine [2,4].

QRISK is a prediction algorithm for CV disease that combines previously studied risk factors, such as age, smoking status, systolic blood pressure and the ratio of total serum cholesterol to high-density lipoprotein cholesterol, together with body mass index, measures of deprivation, ethnicity, family history, diabetes mellitus, rheumatoid arthritis, chronic kidney disease, atrial fibrillation and antihypertensive treatment [5]. As highlighted in the QRISK3 study, some ethnic groups do not have a classic risk map. However, they act as “risk modifiers”, which means that belonging to a specific population is a factor that can “modify” the patient’s risk and treatment decision, especially when the patient’s risk score is borderline. For example, in South Asia, people have higher rates of CVD regardless of other risk factors; therefore, this ethnicity is considered a risk modifier [3,6].

Patients of South Asia (SAP), especially from Bangladesh, are likely to have an earlier and more severe CAD, with twice the prevalence, incidence, hospitalization and death, and a 5-year earlier occurrence of the first CV event but no additional burden of CV risk factors [7,8].

In addition, Bangladeshis tend to have a more aggressive angiographic setting, with nearly double the risk of having two-vessel CAD [9].

Several studies suggest an association between early CAD and a higher risk of mortality associated with ACS in Bangladeshi natives compared with populations from the Western world [10].

Although there are some variances depending on region, the increase in the incidence of cardiac events was greatest in SAP and MENAP and, to a lesser extent, in EEP globally. However, there are some differences depending on the region [11,12].

At the beginning of the 20th century, CAD was responsible for fewer than 10% of all deaths that occurred across the entire world. This figure is estimated to be roughly 30% at the present time, with approximately 80% of the burden falling on developing countries [11].

The Eastern European population (EEP) has a higher mortality rate from CVD and an increased prevalence of risk factors than Central Europe. Compared with the rest of Europe, EEP has one of the highest smoking and excessive alcohol consumption rates, and the incidence of obesity, hypercholesterolemia and type 2 diabetes is increasing [13].

Several explanations have been proposed to explain this discrepancy. EEP has a lower per capita gross domestic product (GDP), which results in reduced healthcare expenditures. However, mortality in EEP is higher than would be predicted by wealth levels, suggesting that ethnicity may be an independent risk factor [14].

In the Middle East and North Africa (MENA) regions, dramatic and rapid lifestyle changes in recent years have led to an increase in the prevalence of CV risk factors and, consequently, a rise in CV mortality at younger ages, influenced by lower incomes [15,16].

In Italy, foreign people represent 8.7% of the population; many citizens come from countries at high risk for CVD, such as South Asia, North Africa and Eastern Europe [17].

These patients are likely to present early and aggressive CAD. Several studies have highlighted the differences between high-risk immigrant groups in Italy, but no study has investigated the exact coronary angiographic findings in this population [18,19].

The Chinese population represents a considerable proportion of foreign residents in Italy. However, a significant amount of their health data is lacking, as they usually use traditional Chinese medicine rather than the national health system.

This retrospective study aims to demonstrate the differential burden of CVD risk factors and specific coronary findings in the above population.

## 2. Materials and Methods

### 2.1. Patient Populations

This study was designed to retrospectively analyze data from our registry at Policlinico Tor Vergata in Rome. From 2016 to 2021, we identified 310 patients from different CV high-risk ethnicities who were referred to our Center for Acute Coronary Syndrome (ACS).

We divided ethnicity into three groups: SAP (Bangladesh, India, Sri Lanka and Pakistan) (*n* = 70), EEP (Romanians, Hungarians, Albanians, Poles, Bulgarians, Moldovans, Czechs, Croatia and Serbia) (*n* = 106) and Middle Eastern and North African patients (MENAP) (Egypt, Morocco, Algeria, Iran, Tunisia, Somalia, Eritrea, Iraq and Libya) (*n* = 44); we included 90 consecutive Italian patients (IP) as a control group.

Our inclusion criteria were: age over 18 years and hospitalization for ACS investigated with angiography. Exclusion criteria were ACS with the absence of coronary angiography.

Nationality, age, sex, CV risk factors known or diagnosed at the time of admission, number of previous CV events, age at onset of first CV event and number of previous procedures CV were obtained from the patient’s medical history.

We analyzed the site and severity of CAD extension using coronary angiography reports. Informed written consent of each patient and ethics committee approval was obtained.

### 2.2. Statistical Analysis

Analyses were performed using the statistical software RStudio version V 1.4.1106 (RStudio, PBC, 2009–2021). All Continuous Variables were firstly tested with Shapiro–Wilk’s Test assessing the normal distribution; then the variables were summarised as mean ± standard deviation (SD) if normality was found; otherwise, they were expressed as a median + interquartile range (IQR). Continuous Variables were used to compare the different ethnical groups using upfront an ANOVA test only after sphericity was ensured by the Mauchly’2 test. When ANOVA resulted in significance, we proceeded with the Student’s t-test if the population distributions were normal; otherwise, a Mann–Whitney U-test was used instead. Categorical variables were compared using the Maenthel–Hanszel Chi-Square for adjusted analyses or Fisher’s exact test whenever the counts were fewer than five individuals, as was the case of the prevalence of three-vessel CAD disease in the 40–50 years old age group. All tests were two-tailed, and *p* < 0.05 was considered statistically significant.

## 3. Results

### 3.1. Study Population

This study retrospectively enrolled 310 patients evaluated with coronary angiography for ACS. Demographic and clinical characteristics and risk factors profile are summarized in Table 1. We compared the correlations between observed CAD and known risk factors in different ethnicities (Table 2).

We find a greater percentage of males in all groups IP 74.4%, SAP 78.6%, EEP 79.2% and MENAP 84.1%.

We observe that IP had a higher average age at the first event than other ethnic groups. The mean age at the first event is 65.4 ± 10.2 years for IP, 49.8 ± 8.5 years for SAP (ReR 30.7%), 51.9 ± 10.2 years for EEP (ReR 26%) and 56.7 ± 11.4 years for MENAP (ReR 15.3%) (*p* < 0.0001) (Table 1).

We did not find any statistical differences concerning dyslipidemia. However, the IP group includes a significative prevalence of hypertension (Table 2). There are no statistically significant differences between IP and SAP for former or current smokers; however, these two groups show a higher prevalence than MENAP and EPP (Table 2).

We found no statistically significant differences in diabetes prevalence between IP and SAP; however, EEP and MENAP have a significative lower prevalence (Table 2).

We observe a significative lower familiarity in the SAP group compared with IP.

### 3.2. ACS Characteristics

Admission population ACS characteristics are reported in Figure 1 and Figure 2.

We find no statistical difference between the four groups in Unstable Angina (UA) (Figure 2a) and Non-ST Elevation Myocardial Infarction (NSTEMI) (Figure 2b) presentation rates. We observe a significant prevalence of ST Elevation Myocardial Infarction (STEMI) (Figure 2c) presentation in EEP and MENAP compared with IP (Figure 2c); otherwise, no difference is found between IP and SAP.

We find an equal anatomical distribution of coronary disease between IP and MENAP. However, we observe in SAP a significant association between the left main artery disease (LM) (*p* = 0.026) and left anterior descending artery disease (LAD) (*p* = 0.033) compared to IP (Table 3). In addition, we observe a higher prevalence of circumflex (Cx) CAD in EEP (*p* = 0.026) compared to IP (Figure 3). 

Notably, comparing the SAP group with all other ethnicities (IP + EEP + MENA), in the younger target age between 40–50 years old, we observe that SAP had a nine-fold higher risk of being affected by three-vessel CAD (OR 9.2; *p* = 0.002, 95% CI 1.8–93.1) (Figure 4).

## 4. Discussion

Our study validated the prior research indicating an early occurrence of CV events in the above high-risk ethnic group [7,8,13,15,16].

SAP is the population with the earliest first CV event, almost 15 years ahead of Italian patients (49.8 ± 8.5 years vs. 65.4 ± 10.2 years).

Contrary to the widespread misconception and the results of previous studies that explained the prevalence of CAD in high-risk groups by a more significant number of risk factors [13,15,16], we found that SAP, MENAP and EEP suffered less from hypertension. Compared to other populations, the prevalence of hypertension was significantly higher among IPs. In addition, a number of studies have demonstrated an increase in the prevalence of overweight and obesity among IPs over the past few decades. Therefore, we hypothesize that these results are attributable to IP’s sedentary lifestyle and a different socioeconomic status. Consequently, this may also account for the high prevalence of hypertension in IP.

In addition, EEP and MENAP had fewer diabetes and smoking habits. However, when we evaluated the sample according to the absolute number of risk factors, IPs were almost three times more likely than EEP to have more than two risk factors (Table 1 and Table 2) with high statistical significance.

EEP had a higher prevalence of STEMI compared with IP, which explains the higher mortality rate from CVD in this population. In the EEP, we found no significant differences in the localization of the coronary lesions, except for the Cx, which is more expressed in the IP (*p* = 0.026).

Statistical analysis of coronary angiography revealed a prevalence of three-vessel disease in SAP, confirmed by the literature [8]. Our data show a higher prevalence of CAD at LM and LAD in SAP, a known predictor of morbidity and mortality. Patients with LM occlusion are at an elevated risk of death due to its anatomic features and also because LM is responsible for supplying blood to 75–100% of the myocardium. Indeed, the LMCA was historically known as the “artery of sudden death” [20,21].

Studies comparing Caucasians with native and expatriate Indians reached similar conclusions. Specifically, these studies showed that compared with Caucasians, Indians were younger at the time of ACS, had lower body mass index, were more likely to have hypertension, were less likely to smoke and had a lower family history of premature CAD [22].

In addition, we found a significantly higher prevalence of trivascular CAD, particularly in a younger age group (40–50 years) in SAP compared with other ethnic groups.

Studies evaluating SAP, especially the Bangladeshi ethnic group, came to a similar conclusion and justified a more than three times higher probability of having a three-vessel CAD [7]. The Study of Health Assessment and Risk in Ethnic Groups (SHARE) is a Canadian population-based study by Anand et al., which investigated three ethnic groups in Canada: SAP, Chinese and EEP. SAs have a 4.5-fold higher risk of developing CAD compared to Caucasians, even after accounting for traditional and novel risk factors [23].

Some research indicates an association between Bangladeshi ethnicity and early CAD and higher mortality risk associated with myocardial infarction. As discussed in a review by Yusuf S. et al., SAP (in the UK and Canada) did not have higher rates of smoking, hypertension, or elevated cholesterol compared with Europeans but did have higher rates of CAD [24].

Recent studies in the USA, among SAP, confirmed the predisposition to early CAD and severe coronary angiographic two- or three-vessel disease [7].

The possible explanations for this concept are multifactorial, including non-modifiable or modifiable risk factors. Several studies demonstrate associations between modifiable risk factors that may cause CAD, such as hypertension or hyperlipidemia. However, additional research is needed on SAP with specific gene polymorphisms that cause CAD and subsequent extensive double or triple CAD.

Because only 1.1% of all deaths are reported to a central agency, data on CVD mortality in MENA countries are limited. In addition, mortality rates for CVD in MENA countries may be higher than in Western countries due to a lack of access to health care [24].

Our data show an increased rate of STEMI presentation in MENAP compared with IP, but no significant differences in coronary lesion extension were found.

There are substantial disparities in mortality rates between European populations. According to EUROSTAT data, the mortality rate due to CVD is four times higher in EEP than in IP [25].

A large proportion of smokers and excessive alcohol use, as well as a diet rich in saturated fats and poor social conditions, contribute to the prevalence of CVD in EEP [26].

Research has found numerous explanations for why IPs are relatively well protected from CVD.

The high consumption of monounsaturated fats, such as olive oil, avocado, nuts, hazelnuts, almonds, pistachios and plant foods, which are known to have antioxidants properties, might be the reason for the low rates of CVD in Italy [26]. Clinical studies have shown that oxidative stress can increase the levels of reactive oxygen species, reducing the production of antioxidant compounds. In addition, increased oxidative stress has been linked to atherosclerosis pathogenesis, especially in plaque destabilization. Studies suggest that in the early stages of the disease, the antioxidant enzyme system is homeostatically upregulated in response to increased free radicals to prevent vascular damage [27].

The main perks of the Mediterranean diet reside in its synergy among various cardio-protective nutrients and foods [28]. This diet has been shown to reduce or even prevent the burden of CV disease and associated risk factors, such as diabetes and obesity, colon cancer, breast cancer, asthma, erectile dysfunction, depression and cognitive decline. Additionally, this diet has been shown to improve primary CV diseases, such as death and events, as well as surrogates for CV disease, including the waist-to-hip ratio, lipids, and inflammatory markers in both observational and randomized controlled trials [29].

Therefore, numerous theories have emerged from studies aimed at finding out why IP remains comparatively well-protected from CVD.

This study suggests that EEP, SAP and MENAP are independent risk factors for CAD and are not related, at least in part, to a supposedly unhealthy lifestyle.

This is even more likely because the sample was collected in an Italian hospital point of reference for emigrant communities; it is reasonable to assume that the patients led a different lifestyle than their compatriots in their country of origin, as demonstrated by several sociological studies. Migrant people tend to change their eating habits once they leave their homeland, consuming a mixed diet of both traditional food from their origin country and typical food from the host country [30,31].

It is not yet known what causes CAD to develop at an earlier age and with greater severity. However, earlier research conducted on children of patients who had early myocardial infarction found that the children had increased platelet reactivity and endothelial dysfunction. This finding suggests that there may be a potential route for future prospective research in these high-risk groups [32].

To the best of our knowledge, this is the first time that a relationship between ethnicity and lesion site has been revealed, which is even more crucial considering that the lesions responsible for the disease are located on the main vessels. These data and a higher prevalence of ACS may explain the higher CV mortality in these high-risk groups. This study suggests that the SAP CAD phenotype is angiographically extensive and has an early onset but is unrelated to excessive CV risk factors expression.

Tor Vergata Hospital is based in Rome and is one of the main centers for CV disease in the region of Latium. Even if Latium is the second most represented region for resident immigrants, our sample does not reflect the immigrant population in Italy because larger proportions are missing, especially Chinese. This could be due to specific cardioprotective factors or, as mentioned above, a general lack of confidence in the Italian NHS [33].

This study has several limitations: It is an observational, retrospective and single-center study. The sample size was relatively small, and there was heterogeneity in CAD prevalence among different ethnicities. Further studies are needed to clarify the topic better.

Although our sample size was limited, our findings confirmed the results of many previous robust epidemiological studies, particularly regarding the prevalence of CV risk factors. The results regarding coronary phenotype were consistent with the few previously published studies on this topic, which we expect to increase in number.

## 5. Conclusions

Our data suggest the existence of a potential coronary phenotype in different ethnicities, particularly SAP, and underplays the prevalence of CV risk factors in other high-risk groups, confirming the importance of genetics in these populations. Further, more extensive and prospective studies are required to confirm the data and identify the rationale according to the principles of modern tailored medicine.

## Figures and Tables

**Figure 1 jpm-13-00882-f001:**
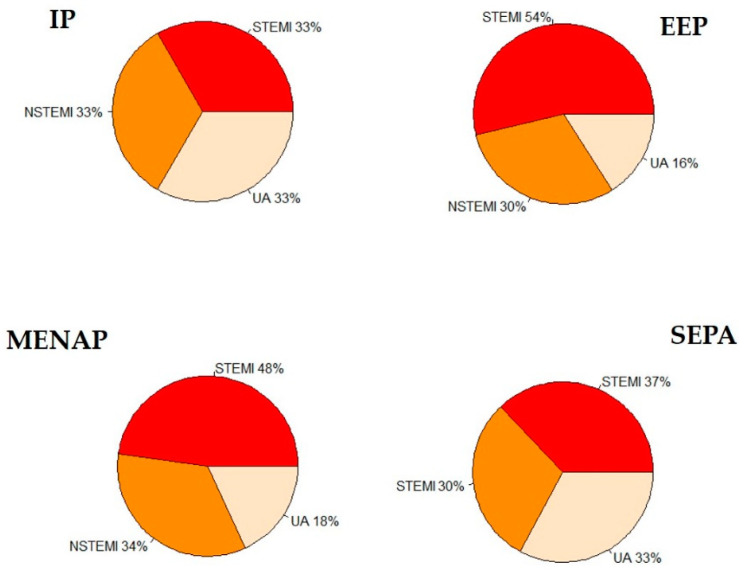
Admission Acute Coronary Syndrome (ACS) population characteristics. IP = Italian patients; EEP = East European patients; MENAP = Middle Eastern and North African patients; SAP = South Asian patients; NSTEMI: Non-ST Elevation Myocardial Infarction; STEMI: ST Elevation Myocardial Infarction; and UA: Unstable Angina.

**Figure 2 jpm-13-00882-f002:**
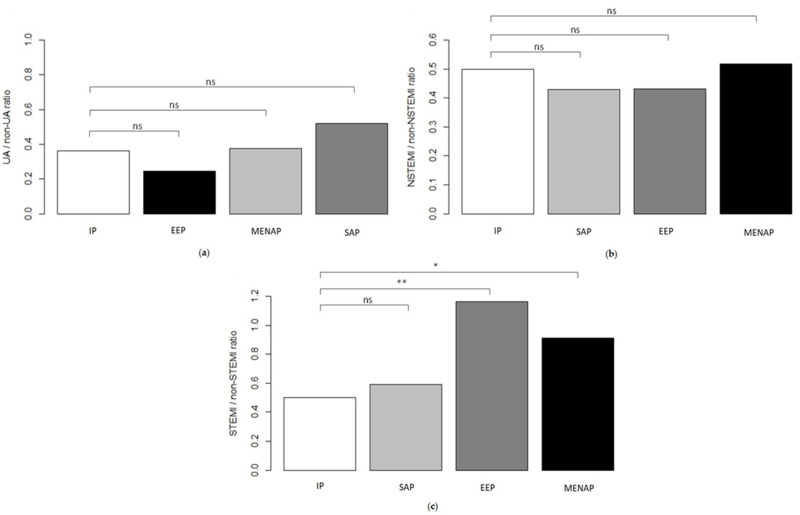
Ratio Admission Acute Coronary Syndrome (ACS) population characteristics. IP = Italian patients; EEP = East European patients; MENAP = Middle Eastern and North African patients; SAP = South Asian patients; NSTEMI: Non-ST Elevation Myocardial Infarction; STEMI: ST Elevation Myocardial Infarction; and UA: Unstable Angina. (**a**) UA/non-UA ratio. (**b**) NSTEMI/non NSTEMI ratio. (**c**) STEMI/non STEMI ratio. ns = not significant (* *p* < 0.05 and ** *p* < 0.001).

**Figure 3 jpm-13-00882-f003:**
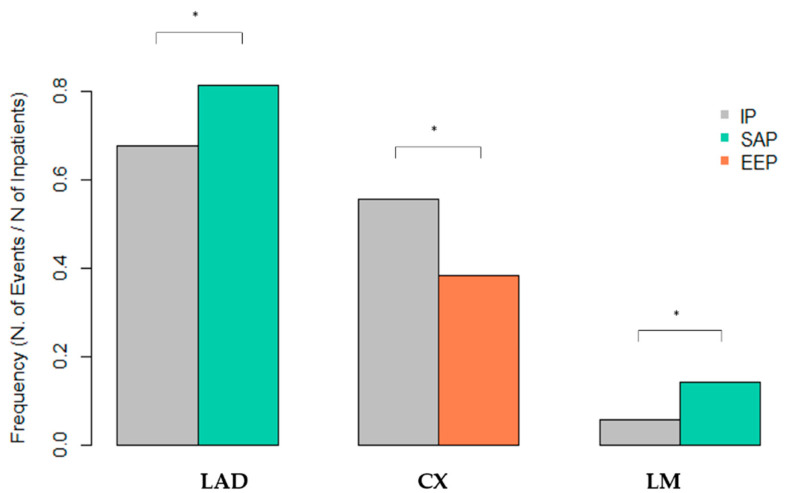
Significant extent of CAD. LAD: left anterior descending artery; Cx: Circumflex artery; LM: left main; IP = Italian patients; SAP = South Asian patients; EEP = East European patients; and CAD: coronary artery disease. (* *p* < 0.05).

**Figure 4 jpm-13-00882-f004:**
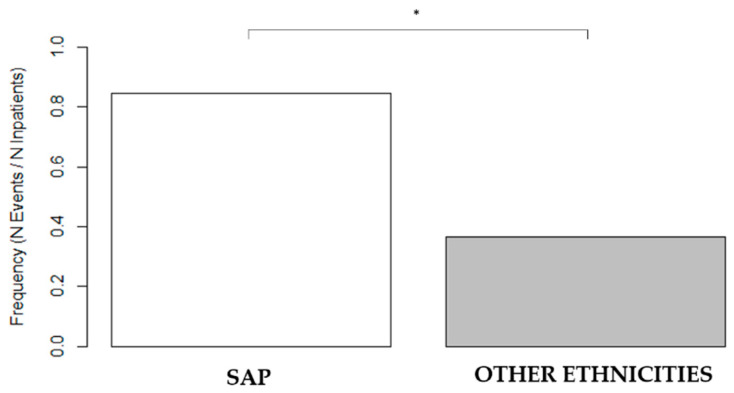
Prevalence of three-vessel CAD in a younger age group, 40–50 years old, in the SAP population compared with other ethnicities. Frequencies (N events/N Inpatients): SAP 11/13 = 0.85; other ethnicities 22/60 = 0.37. SAP = South Asian patients and CAD = coronary artery disease (* *p* < 0.05).

**Table 1 jpm-13-00882-t001:** Demographic characteristics.

Ethnicities		IP (*n* = 90)	SAP (*n* = 70)	EEP (*n* = 106)	MENAP (*n* = 44)	*p*-Value
Male		67 (74,4%)	55 (78,6%)	84 (79.2)	37 (84.1)	< 0.0001
Mean age at the first event (ReR)		65.4 ± 10.2	49.8 ± 8.5	51.9 ± 10.2	56.7 ± 11.4	< 0.0001
Dyslipidemia		81 (90.9%)	61 (87.1%)	97 (91.5%)	38 (86.4%)	0.72
Hypertension		81 (89.9%)	45 (64.3%)	77 (72.6%)	32 (72.7%)	0.003
Diabetes		55 (61.6%)	43 (61.4%)	22 (20.8%)	19 (43.2%)	< 0.0001
Current or former smokers		72 (80%)	40 (44%)	61 (26%)	21 (23%)	0.0005
Familiarity (IQR)		35 (38.9%)	10 (14.3%)	32 (30.2%)	12 (27.3%)	0.004
Multiple risk factors	4 risk factors	16 (18.2%)	10 (14.3%)	7 (6.6%)	8 (18.2%)	0.032
	3 risk factors	44 (48.5%)	28 (40.0%)	35 (33.0%)	13 (29.5%)	
	2 risk factors	23 (25.3%)	24 (34.3%)	46 (43.4%)	18 (40.9%)	
	1 risk factor	7 (8.1%)	8 (11.4%)	18 (17.0%)	5 (11.4%)	

Data are expressed as number (*n*), mean ± standard deviation or percentages (%). ReR: Relative Reduction; IP = Italian patients; SAP = South Asian patients; EEP = East European patients; and MENAP = Middle Eastern and North African patients.

**Table 2 jpm-13-00882-t002:** Adjusted * Odds Ratios (95% confidence interval) for cardiovascular risk factors.

Ethnicities		SAP (*n* = 70)	EEP (*n* = 106)	MENAP (*n* = 44)
Dislipidemia		[OR 0.68; CI: 0.2–2.0] *p*-value: 0.60	[OR 1.1; CI: 0.4–3.2] *p*-value: 1	[OR 0.64; CI: 0.2–2.3] *p*-value: 0.39
Hypertension		[OR 0.28; CI: 0.1–0.5] *p*-value: 0.001	[OR 0.3; CI: 0.1–0.7] *p*-value: 0.003	[OR 0.3; CI: 0.1–0.8] *p*-value: 0.018
Diabetes		[OR 1; CI: 0.5–2.0] *p*-value: 1	[OR 0.16; CI: 0.1–0.3] *p*-value < 0.0001	[OR 0.48; CI: 0.2–0.9] *p*-value: 0.04
Current or former smokers		[OR 1.22; CI: 0.6–2.5] *p*-value: 0.67	[OR 0.4; CI: 0.2–1.0] *p*-value: 0.008	[OR 0.37; CI: 0.1–0.9] *p*-value: 0.03
Familiarity (IQR)		[OR 0.24; CI: 0.1–0.5] *p*-value: 0.0003	[OR 0.61; CI: 0.3–1.1] *p*-value: 0.13	[OR 0.53; CI: 0.2–1.2] *p*-value: 0.15
Multiple risk factors	4 risk factors	[OR 0.75; CI: 0.3–1.9] *p*-value: 0.54	[OR 0.3; CI: 0.1–0.9] *p*-value: 0.2	[OR 1; CI: 0.3–2.7] *p*-value: 1
	3 risk factors	[OR 0.71; CI: 0.4–1.4] *p*-value: 0.35	[OR 0.52; CI: 0.29 0.96] *p*-value: 0.03	[OR 0.45; CI: 0.2–1.0] *p*-value: 0.034
	2 risk factors	[OR 1.54; CI: 0.7–3.2] *p*-value: 0.23	[OR 2.26; CI: 1.2–4.3] *p*-value: 0.01	[OR 2; CI: 0.9–4.6] *p*-value: 0.06
	1 risk factor	[OR 1.32; CI: 0.4–4.2] *p*-value: 0.80	[OR 2.3; CI: 0.9–6.5] *p*-value: 0.06	[OR 1.45; CI: 0.4–5.4] *p*-value: 0.53

OR: Odds Ratio; CI: confidence interval; IQR: interquartile range. SAP: South Asian patients; EEP: East European patients; and MENAP: Middle Eastern and North African patients. * The values were adjusted through the use of the Maenthel–Haenszel Chi-Square Test.

**Table 3 jpm-13-00882-t003:** Presentation of the extent of CAD in the four groups.

	LAD		Cx		RCA		LM		3VD		2VD	
	R.F. (%)	*p*-Value	R.F. (%)	*p*-Value	R.F. (%)	*p*-Value	R.F. (%)	*p*-Value	R.F. (%)	*p*-Value	R.F. (%)	*p*-Value
IP	66.7	- *	55.6	- *	47.8	- *	5.6	- *	27.8	- *	24.4	- *
SAP	82.9	0.033	52.9	0.86	58.6	0.23	15.7	0.026	30.0	0.90	35.7	0.17
EEP	75.5	0.300	38.7	0.026	45.3	0.84	4.7	1.00	22.9	0.53	23.6	1.00
MENAP	68.2	1.00	50.0	0.67	50.0	0.95	7.0	0.70	22.7	0.68	31.8	0.37

LAD: left anterior descending artery; Cx: Circumflex artery; RCA: right coronary artery; LM: left main; 3VD: three-vessels disease; 2BV: two-vessels disease; and RF: Relative Frequency. * IP group was used as a control arm to compare differences with other ethnicities.

## Data Availability

Data supporting reported results can be provided upon request of the corresponding author.

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
