# Peer review of "Prevalence of Cardiovascular Risk Factors and Coronary Angiographic Findings in High-Risk Immigrant Communities in Italy"

_jpm, 2023, doi:10.3390/jpm13060882_

Round 1

Reviewer 1 Report

I revised an interesting manuscript draft entitled “Prevalence of Cardiovascular Risk Factors and Coronary Angiographic Findings in High-Risk Immigrant Communities in Italy”.

In their retrospective study, the authors aimed to find the cardiovascular risk (CV) factors and specific coronary findings in high-risk immigrant groups in 220 different ethical grups medical records of coronary angiographies and compare them with medical records of 90 Italian patients.

The results obtained suggest the existence of a potential coronary phenotype in several ethnicities, especially in south Asian patients, and understate the frequency of CV risk factors in other high-risk groups, supporting the role of a genetic influence in these communities.

Please correct the text according my suggestions.

Line 18: Please explain the used abbreviation ACS by its full name, acute coronary syndrome.

Line 19: Please explain the used abbreviation RR by its full name (relative risk?).

Line 43: The cited reference numbers have to be in bracket before the full stop of the end of the sentence.  “...and Ukraine [2-4].” Please correct them in the full text.

Line 44: Please explain the used abbreviation QRISK3 in its first order of appearance. After that you may use its abbreviated name.

Line 81: Please explain the used abbreviation ACS (acute coronary syndrome) independently that you explained it in the abstract. After that you may use its abbreviated name ACS.

Line 113: “We found a significant proportions...”. What is and what kind of significant proportion? Does 74.4% and 78.6% in IP and SAP are significant proportions? Do you mean on significant difference between in proportion? Where is a p-value for that significancy?

  Table 1: Please insert an explanation in the bottom of the table: “The results are expressed as number (N) and percent (%), and mean ± standard deviation. Please provide a sixth column beside column “MENAP” with title P and provide p of statistical difference among IP, SAP, EEP and MENAP groups. You may use a test for multiple group comparison (multiple group comparison tests such as ANOVA.     Table 2: In the sixth row named “Familiarity (IQR) you provide results for odds ratio (OR) and confidence interval (QI). Where is and what is IQR? Please replace “p value: 0.54” with “p = 0.54”. It is clear and known that “p” represents “p value”. Please delete all “p value:” in the text. Please explain the used abbreviations SAP, EEP, MENAP by their full names in the bottom of the table.  

Table 3: In the bottom of the table, please provide an explanation what star (asterisk) mean (*)?

Figure 2: In the statistics, significance is either there or not. It is defined by P (P < 0.05). Terms like hardly, or slightly significant are not allowed. Delete the asterisks (*,** and ns)  and put the correct value of p. Use of abbreviations in titles of figures and tables is not allowed. Explain the abbreviations used in the title (ACs, UA, NSTEMI, STEMI, IP, SAP, EEP, MENAP) and below the diagrams at the bottom of the table.

Please do the same in the Figure 1.

Figure 4: Please provide a p value for statistical difference between SAP and other ethnicities groups, instead asterisk (*).

There is no need to explain the used statistical method in the bottom in the figures and the tables. Please delete the sentence “Due to low absolute frequencies a fisher’s exact test was performed.”. Do the same in table 2. The used statistical method have to be explained in part : “Statistical analysis”.

Тhe manuscript is well conceived and the results are well processed retrospectively. Apart from the small corrections that are needed in the presentation of results, everything else is understandable. The discussion is based on the present results and is accordingly compared with results from other studies. The conclusions are well presented and derive from the results of the study. If all my remarks are corrected, the paper will be significantly improved and may be ready for publication. The authors have to respond on my review on every line of suggestions, in this way:   Reviewer: Line 18: Please explain the used abbreviation ACS by its full name, acute coronary syndrome. Authors: Dear reviewer, thank you for your suggestion. I am agreed (or not agreed, and why) with it. I explained the used abbreviation ACS with its full name acute coronary syndrome and this explanation is inserted in the manuscript text on line (insert the line number) with red highlight.

Reviewer 2 Report

The authors evaluated the risk factors and coronary involvement among different ethnicities in a single center. Even though there is no methodical flaws significance of the work is limited given the small sample, retrospective and inherent limitations due to the nature of the study. The value of the study may be guarded due to the above-mentioned limitations and need to be cautious while generalizing the findings. The different risk factors observed among ethnicities observed and the proportion of the ACS may differ in more robust epidemiological studies. 

These limitations need more elaboration and need more robust discussion.

Reviewer 3 Report

ligne 231= data 

This  study describe  the differential burden of CVD risk factors and specific coronary findings in immigrant communities and to compare with italian population . Understanding of cardiovascular disease risk factors among this population is utmosts of importance because the total variability in CAD risk factors could not been explained the highest incidence of ACS. It is important some unidentified risk factors. this studi suggest that environmental influence and lifestyle change that took place after migration. These hypothesis neds to be explored further. Only a few data has been published in this area 

Round 2

Reviewer 2 Report

In the abstract " We analyzed the medical records of 220 patients from the high-risk ethnic groups described above referred for ACS and compared them with 90 IP between 2016 and 2021." is redundant rest is otherwise appropriately revised. with proofreading and minor revision will be appropriate to move forward with the publication!